# The Annexin A2/S100A10 Complex: The Mutualistic Symbiosis of Two Distinct Proteins

**DOI:** 10.3390/biom11121849

**Published:** 2021-12-09

**Authors:** Alamelu Bharadwaj, Emma Kempster, David Morton Waisman

**Affiliations:** 1Department of Pathology, Faculty of Medicine, Dalhousie University, Sir Charles Tupper Medical Building, Halifax, NS B3H 1X5, Canada; Alamelu.Bharadwaj@Dal.ca (A.B.); em820143@dal.ca (E.K.); 2Department of Biochemistry and Molecular Biology, Faculty of Medicine, Dalhousie University, Halifax, NS B3H 1X5, Canada

**Keywords:** S100A10, annexin A2, plasminogen, plasmin, ion channels

## Abstract

Mutualistic symbiosis refers to the symbiotic relationship between individuals of different species in which both individuals benefit from the association. S100A10, a member of the S100 family of Ca^2+^-binding proteins, exists as a tight dimer and binds two annexin A2 molecules. This association forms the annexin A2/S100A10 complex known as AIIt, and modifies the distinct functions of both proteins. Annexin A2 is a Ca^2+^-binding protein that binds F-actin, phospholipid, RNA, and specific polysaccharides such as heparin. S100A10 does not bind Ca^2+^, but binds tPA, plasminogen, certain plasma membrane ion channels, neurotransmitter receptors, and the structural scaffold protein, AHNAK. S100A10 relies on annexin A2 for its intracellular survival: in the absence of annexin A2, it is rapidly destroyed by ubiquitin-dependent and independent proteasomal degradation. Annexin A2 requires S100A10 to increase its affinity for Ca^2+^, facilitating its participation in Ca^2+^-dependent processes such as membrane binding. S100A10 binds tissue plasminogen activator and plasminogen, and promotes plasminogen activation to plasmin, which is a process stimulated by annexin A2. In contrast, annexin A2 acts as a plasmin reductase and facilitates the autoproteolytic destruction of plasmin. This review examines the relationship between annexin A2 and S100A10, and how their mutualistic symbiosis affects the function of both proteins.

## 1. Introduction

The nomenclature for the S100 family of Ca^2+^-binding proteins is derived from the observation that these proteins are soluble in a saturated ammonium sulfate solution at a neutral pH. The S100 family of proteins consists of about twenty 10–12 kDa, Ca^2+^-binding proteins that are expressed only in vertebrates and consist of S100A1–S100A18, S100B, S100G, S100P, S100Z, as well as members with an N-terminal S100 domain fused to additional C-terminal sequences, such as repetin [1], trichohyalin and filaggrin [2,3,4]. Of the twenty-five human *S100* genes, 19 are located within chromosome 1q21. Other members (*S100A11*, *S100B*, *S100G*, *S100P*, and *S100Z*) map to different chromosome regions [5]. Each S100 protein has a unique expression and distribution profile amongst different tissues and cell types. The S100 proteins exist as anti-parallel homo- and heterodimers, with each monomer consisting of two helix-loop-helix EF-hand motif type, Ca^2+^-binding domains (EF-1 and EF-2) connected by a hinge region. The nomenclature for the EF-hand domain originally comes from the EF-hand motif first identified in the Ca^2+^-binding protein, parvalbumin, which is composed of two alpha-helices “E” and “F” connected by an intermediate loop of 12 residues binding Ca^2+^ [6]. Specifically, the S100 proteins possess an S100-specific EF hand at the N-terminus and a canonical EF hand at the C-terminus. In all of the S100 proteins except S100A10, the EF hands bind Ca^2+^. These EF hands are not identical; the carboxyl-terminal EF hand Ca^2+^-binding loop contains the classical 12-amino-acid long Ca^2+^-binding motif that is common to all EF hand Ca^2+^-binding proteins, such as parvalbumin and calmodulin. In this motif, Ca^2+^-binding occurs via acidic side chains that comprise the sequence DXDGDGTIXXXE. However, the N-terminal EF hand motif is a 14-amino-acid long Ca^2+^-binding loop which is characteristic of all S100 proteins, and is referred to as the S100-specific or pseudo-EF domain. This motif binds Ca^2+^ via backbone carbonyl groups and one carboxylate side group contributed by glutamic acid (reviewed in [2,7,8]). All of the S100 proteins bind calcium, except S100A10, which has lost its ability to bind calcium due to substitutions in its calcium-binding loop, but retains the “active” conformation of a calcium-bound S100 protein [9]. S100A10 lost its ability to bind calcium due to substitutions in its calcium-binding loop, but retains the activated structure of a calcium-bound S100 protein [10] (Figure 1).

The S100 proteins S100A1, S100A4, S100A6, S100A10, S100A11, S100A12, and S100B can interact with several annexin (ANX) proteins. The annexins are a family of proteins that bind multiple Ca^2+^ ions via Ca^2+^-binding sites that are not EF-hand motifs, but are referred to as type II and type III Ca^2+^-binding sites [11]. The X-ray crystallographic analysis of ANXA2 established a single type II Ca^2+^-binding site in each of the second, third, and fourth domains, and two type III Ca^2+^-binding sites in the first domain of the protein. The characteristic feature of all annexins is the presence of several copies of an annexin repeat, which is defined as an approximately 70-residue-long conserved structural element that is required for the Ca^2+^ and membrane-binding function of the annexins. The presence of such annexin repeats combined with the capability of a protein to interact with the negatively charged phospholipids in a Ca^2+^-dependent manner is the major criterion defining the family of annexins. There are 12 human annexins (ANXA1-ANXA11 and ANXA13). ANXA1, ANXA2, ANXA5, ANXA6, and ANXA11 interact with S100 proteins [12]. S100A1, S100A4, S100A6, S100A10, S100A11, S100A12, and S100B can interact with annexins. Interestingly, the S100 proteins and annexins possess the ability to form multiple complexes. For example, ANXA2 is found in complexes with S100A4, S100A6, S100A10, and S100A11 [13]. However, the complex formation is much more favorable in the case of S100A10 (Kd of 13 nM) compared to S100A4 (Kd of 5 uM) [14]. Furthermore, although both S100A4 and S100A10 promote plasminogen activation [15,16], the activation of plasminogen by S100A4 but not S100A10 is Ca^2+^-dependent. The structural and functional aspects of various S100-annexin complexes were discussed in dedicated reviews [13,17,18,19], and only the ANXA2/S100A10 complex will be discussed in this review.

The S100 proteins are typically dimeric, and have two specialized binding regions that are either identical or slightly different in the case of homo- and heterodimers, respectively [20]. These binding regions act as anchor points for their ligands, but are not surface exposed in the absence of Ca^2+^. They become accessible when Ca^2+^ binds to the EF-hands of the S100 proteins and induces a conformational rearrangement in them. These binding regions in S100A10 are surface exposed in the presence or absence of Ca^2+^. Between the binding pockets of the S100 proteins there exists a shallow groove. This region is located at the dimeric interface. The shallow groove is less specific with regard to the binding partners that interact with it (reviewed in [21]. Two identical binding proteins can occupy these sites, as exemplified by the ANXA2–S100A10 complex [22]. In contrast, in some cases a single binding sequence utilizes both sites [21,23] and can also utilize the groove to various extents [21]. For example, two molecules of S100A4 bind to a single ANXA2 molecule, i.e., one molecule of ANXA2 occupies both binding pockets of the S100A4 dimer, whereas only one molecule of ANXA2 occupies one site on each S100A10 molecule in the dimer. It has been suggested that in order to be able to bind a wide variety of ligands without optimal binding motifs, S100 proteins may have developed relatively permissive binding sites, leading to more flexible interactions [17,21].

S100A10 was initially identified in a complex with ANXA2. This complex, for which we have coined the term AIIt, was discovered by two different groups. One group utilized two-dimensional polyacrylamide gel electrophoresis to examine the effects of viral transformation on the phosphorylation pattern of cellular proteins. They identified the prominent phosphorylated protein as a 36-kDa protein that underwent phosphorylation on tyrosine residues after the transformation of cells by the Rous Sarcoma Virus (RSV) [24,25,26]. Subsequently, it was shown that ANXA2 existed in a complex with an 8–10K binding partner (S100A10) in the form of a heterotetramer [27]. Independently, other laboratories identified an F-actin-binding protein, called Protein I, as a molecular weight 85-kDa protein that consisted of two copies of a 36-K subunit and two copies of a 10-kDa subunit [9,28,29]. Subsequent studies detailed the tissue distribution of S100A10. It was shown that this protein is widely distributed in various tissues, such as the gastrointestinal tract, brain, heart, liver, kidney, lung, spleen, testes, aorta, and thymus [30]. Studies with cultured cells have identified the presence of S100A10 in endothelial cells, cardiomyocytes, monocytes, macrophages, lymphocytes, neurons, and reactive astrocytes (reviewed in [8,31]. 

The structure of ANXA2 comprises two distinct regions (Figure 2). The N-terminal domain (residues 2–33) is an intrinsically disordered region. The first twelve residues form an amphipathic α-helix, and the hydrophobic face of this helix—specifically Val3, Ile6, Leu7, and Leu10 of S100A10—interacts with a hydrophobic surface on the S100A10 dimer [22,28,32,33]. These four hydrophobic amino acids of the N-terminal domain of ANXA2 form seven points of contact with helix H-I of one monomer, two points of contact with the hinge region, and nine points of contact with helix H-IV of the other monomer, for a total of nineteen points of contact between ANXA2 and S100A10 (reviewed in [8,17]. This region also bears two important phosphorylation sites, Tyr23 and Ser25, which are targets of SRC kinase and protein kinase C, respectively [9,27,34].

The conserved C-terminal core domain (residues 34–339) contains four annexin repeats. This domain has a convex side which is responsible for calcium and membrane-binding, whereas the concave surface anchors the N-terminal domain and the binding site for S100A10. The C-terminal core domain also interacts with many biomolecules, including RNA [35,36,37,38], phospholipid [39,40,41], heparin [42], and F-actin [39,43,44,45]. 

As will be discussed in this review, S100A10 has been shown to possess many intracellular and extracellular functions. The S100A10 knockout mouse [46] has been particularly instructive in highlighting several essential functions for this protein, including roles in depression [31,46], homeostasis [47], and cancer [48,49,50]. The purpose of this review is not to provide an extensive review of the many and diverse functions of S100A10, as this has been covered extensively by others [31,49,51,52,53,54,55,56,57]. Instead, we propose to illustrate how S100A10 and its binding partner, ANXA2, function in a mutualistic symbiotic relationship, and how this relationship governs their biological activity. The mutualistic symbiotic relationship between S100A10 and ANXA2 is summarized in Figure 3.

## 2. Mutualistic Symbiosis

Mutualistic symbiosis refers to the symbiotic relationship between individuals of different species in which both individuals benefit from the association. We propose that this biological relationship between species can be extended to the subunits within a protein complex. As will be discussed in detail, the formation of the ANXA2/S100A10 complex not only regulates the biological activity of each subunit but also confers new biological functions that were not possessed by either subunit. Furthermore, complex formation is also critical for the survival of S100A10 because, in the absence of ANXA2, the protein is rapidly degraded [58]. 

### 2.1. Mutualistic Symbiosis I-Survival of S100A10

S100A10 is undetectable unless ANXA2 is also expressed [58,59,60,61,62]. Thus, S100A10 is an extrinsically disordered protein that is rapidly degraded [63]. The binding of S100A10 to DLC1, however, does not protect S100A10 from degradation [64]. However, ANXA2 competes with DLC1 for binding to S100A10, suggesting that the interaction of S100A10 with ANXA2 or DLC1 results in unique conformations of S100A10. Previous studies demonstrated that the depletion of ANXA2 from cells resulted in the rapid ubiquitylation and proteasomal degradation of S100A10. One of these studies utilized protein overexpression with site-directed S100A10 mutants to propose that the carboxyl-terminal residues, Lys-91 and/or Lys-93, function as the ubiquitylation sites [65]. However, these studies failed to directly confirm the presence of ubiquitylated S100A10 at these sites by mass spectrometric analysis. We have also observed that the forced expression of S100A10 and ubiquitin results in the ubiquitination of S100A10, and we identified, for the first time, that the ubiquitination of S100A10 in forced expression systems occurs on Lys-57 [66]. Our study also showed that the mutagenesis of Lys-57 prevents the ubiquitylation and degradation of overexpressed S100A10. In contrast, another group identified Lys-47 as a site of the succinylation of S100A10, and proposed that this site was also a site of ubiquitylation [67]. Interestingly, Lys-47, Lys-54, and Lys-57 were identified as sites of the ubiquitylation of S100A10 by the mass spectrometric analysis of tissue extracts [68], although these sites were not confirmed by site-directed mutagenesis. It was also not clear why only two of the five tissues examined in that study detected ubiquitinated S100A10. It is interesting to note that intrinsically disordered regions have been identified in S100A10, and that Lys-57 is located in the helix III, which is an intrinsically disordered region [63]. We analyzed the amino acid sequence of the S100A10 of twenty-one vertebrate species, and although Lys-57 is conserved in all of these vertebrate species, Lys-47 is not conserved in chickens, xenopus or pufferfish.

Ubiquitylated proteins are generally degraded by the 26S proteasome, whereas protein degradation by the 20S proteasome does not require prior ubiquitylation [69,70]. Several studies of the ubiquitylation of S100A10 involved the use of lactacystin to show the recovery of the S100A10 levels in ANXA2-depleted cells. These studies then concluded that S100A10 was regulated by ubiquitin-dependent proteasomal degradation. However, lactacystin binds certain catalytic subunits of the 20S proteasome and inhibits it irreversibly. Thus, the proteasomal degradation of proteins mainly occurs by ubiquitin-dependent degradation by the 26S proteasome, as well as ubiquitin-independent degradation by the 20S proteasome. Lactacystin inhibits both of these mechanisms by targeting the catalytic β-subunit of the 20S proteasome core. Therefore, lactacystin is an inhibitor of ubiquitin-dependent and -independent proteasomal degradation [71,72]. From these studies, we concluded that ANXA2 stabilizes S100A10 and prevents its ubiquitin-dependent and -independent proteasomal degradation. 

### 2.2. Mutualistic Symbiosis II-Regulation of the Biological Functions of ANXA2 and S100A10

The mutualistic symbiotic role of S100A10 binding is apparent from studies of the regulation of F-actin bundling. Individually, S100A10 and ANXA2 are incapable of the significant bundling of F-actin; however, when ANXA2 bound S100A10, the complex facilitated the Ca^2+^-dependent formation of anisotropic F-actin bundles [43]. Because ANXA2 is an F-actin-binding protein, whereas S100A10 does not interact with F-actin, this type of mutualistic symbiosis involves the modification of the function of one subunit by the other.

AIIt-dependent F-actin bundles participate in the spatial organization of the plasma membrane, providing active sites for secretory granule docking and exocytotic fusion. For example, it has been shown that when chromaffin cells are stimulated with nicotine, the F-actin-bundling activity of ANXA2 promotes the formation of the monosialotetrahexosylganglioside and GM1-enriched microdomains at the plasma membrane, increases the number of morphologically docked granules at the plasma membrane, and controls the number of individual exocytotic events [73]. Conversely, when an ANXA2 mutant with impaired F-actin filament–bundling activity [44] was expressed, the formation of plasma membrane lipid microdomains and the number of exocytotic events were decreased, and the fusion kinetics were slower [73]. Based on these studies, the authors concluded that AIIt-induced F-actin bundling is essential for the generation of active exocytotic sites.

Chasserot-Golaz’s group proposed a model for the involvement of AIIt in neurosecretion that incorporates both the F-actin bundling and the lipid sequestration properties of AIIt [74]. In unstimulated adrenergic cells, ANXA2 is cytosolic, S100A10 is bound to vesicle-associated membrane protein 2 (VAMP2) at the plasma membrane, and the membrane lipids PIP2 and GM1 are randomly distributed in the plasma membrane. However, upon stimulation with secretagogues, ANXA2 translocates to the plasma membrane in proximity to N-ethylmaleimide-sensitive factor attachment protein receptor (SNARE) complexes; this redistribution of ANXA2 is mediated by S100A10 [75]. The newly formed AIIt forms specific lipid microdomains. AIIt then bundles F-actin, which constrains the lateral mobility of the newly formed lipid domains, thereby increasing their stability. This compartmentalization of the plasma membrane serves to spatially and temporally organize F-actin bundles, which contribute to the docking machinery and thereby accelerate the vesicle fusion proteins required for secretory granule docking and fusion, likely playing a role in both the speed and accuracy of the exocytotic fusion process.

The biochemical properties of the monomeric ANXA2 and ANXA2 present in the AIIt complex are distinct. ANXA2 typically exists as a monomer predominantly localized to the cytosol and on early endosomes [76], or as a heterotetrameric complex—AIIt—that is found in the subplasmalemmal region [77]. Several laboratories have reported that the binding of S100A10 to ANXA2 induces a conformational change in ANXA2, which influences its interaction with several of its ligands. For example, S100A10 binding has a stimulatory effect on phospholipid binding by ANXA2, and the phospholipid requirement for 50% binding is 10-fold lower for the complex than ANXA2 [78]. We initially investigated the consequence of complex formation on the biological properties of the individual subunits by expressing and purifying recombinant ANXA2 and S100A10, and then combining the subunits and isolating AIIt. Previously, it was reported that only the ANXA2/S100A10 complex—and not monomeric ANXA2—could aggregate membrane vesicles at submicromolar Ca^2+^ concentrations [79,80]. We found that the binding of recombinant S100A10 to recombinant ANXA2 resulted in a decrease in the A0.5 (Ca^2+^) for chromaffin granule aggregation from 0.23 mM for recombinant ANXA2 to 1.0 μM for the recombinant complex. We also found that the binding of recombinant S100A10 to recombinant ANXA2 resulted in a decrease in the A0.5 (Ca^2+^) of phospholipid liposome aggregation from 0.83 μM for recombinant ANXA2 to 0.26 μM for the recombinant complex. In contrast, S100A10 did not interact with chromaffin granules or phospholipid vesicles.

Many studies have shown that ANXA2 and AIIt organize cholesterol-rich lipid rafts [81,82] and link them to cytoskeletal proteins [83,84]. It has also been reported that ANXA2 and AIIt bind to phosphatidylinositol 4,5-bisphosphate (PtdIns(4,5)P2) with high specificity and affinity, and that this activity is linked to the organization of actin at membrane sites that are enriched in PtdIns(4,5)P2 [85,86]. ANXA2 can induce the formation of phosphatidylinositol 4,5-bisphosphate-rich microdomains in the giant unilamellar vesicles (GUVs) of complex lipid composition [87,88]. AIIt has a 10-fold higher affinity for PtdIns(4,5)P2-containing vesicles than ANXA2 at 0.1 mm Ca^2+^, but at 0.05 mM Ca^2+^ ANXA2 did not bind this lipid, suggesting that S100A10 increases the affinity of ANXA2 for PtdIns(4,5)P2, and that the interaction of ANXA2 with PtdIns(4,5)P2 is unlikely to occur physiologically [88]. Therefore, S100A10 confers to ANXA2 the properties of lipid sequestration and F-actin bundling, but does not directly interact with lipid or F-actin. 

Our laboratory originally identified AIIt as a plasminogen receptor, and showed that the S100A10 subunit bound both tissue plasminogen activator (tPA) and plasminogen, and stimulated the tPA and urokinase plasminogen activator (uPA)-dependent conversion of plasminogen to plasmin [16,89]. It has been proposed that ANXA2 does not bind plasminogen. However, a proteolytically processed form, ANXA2_1-307_, is capable of both binding plasminogen and activating tPA-dependent plasminogen activation (reviewed in [8,90]). The issue of whether intact annexin A2 binds plasminogen has been investigated by our group and by Plow’s group. Both groups have clearly stated that annexin A2 does not bind plasminogen or participate in plasminogen activation [16,91,92]. However, because the ANXA2_1-307_ form of ANXA2 has not been detected in cellulo or in vivo, this proposal has been abandoned. Furthermore, although it was originally proposed that ANXA2 bound tPA, recent studies with surface plasmon resonance have failed to reproduce this report [91]. Furthermore, a peptide to the putative tPA-binding site of ANXA2 failed to bind tPA [93]. Rather, it has been shown that ANXA2 stimulates the plasminogen binding and activation activity of S100A10. For example, S100A10 bound t-PA (Kd = 0.45 μM) and plasminogen (Kd = 1.81 μM), whereas AIIt bound t-PA (Kd = 0.68 μM) and plasminogen (Kd = 0.11 μM). When AIIt is incubated with carboxypeptidase B, the carboxyl-terminal lysines of the S100A10 subunit are removed [94]. Under these conditions, both tPA and plasminogen binding to AIIt is lost, and the ability of AIIt to stimulate the tPA-dependent conversion of plasminogen to plasmin is reduced by almost 90% [91,94]. Furthermore, a mutant AIIt composed of the wild-type p36 subunit and the S100A10 subunit deletion mutant possessed about 12% of the wild-type activity [16]. When we compared the ability of the isolated ANXA2 and S100A10 subunits to stimulate t-PA-dependent plasminogen activation, we observed that the recombinant S100A10 subunit stimulated this rate about 46-fold compared to an approximate twofold stimulationby the recombinant ANXA2 subunit. In contrast, the recombinant AIIt stimulated this rate by 77-fold. Furthermore, we observed that when S100A10 formed a complex with a peptide of the first 14 amino acid residues of the N-terminal region of ANXA2, representing the region of ANXA2 that binds S100A10, the activity of S100A10 was stimulated to levels similar to that of AIIt [16]. These studies clearly demonstrate that within AIIt, the ANXA2 subunit stimulates the ability of the S100A10 subunit to stimulate tPA-dependent plasminogen activation.

Plasmin is capable of digesting itself, in a process called autoproteolysis. This process is a bimolecular reaction in which one plasmin molecule attacks another, which results in the proteolysis of the protein and a loss in plasmin activity [95]. We have observed that both subunits of AIIt bind plasmin (ANXA2, 0.78 μM; S100A10, 0.36 μM) [91] and stimulate plasmin autoproteolysis [96]. Not only did AIIt stimulate the ability of plasmin to cleave itself, but AIIt also acted as a plasmin reductase, which allowed the dissociation of the proteolyzed polypeptide chains of plasmin [97,98,99,100]. We proposed a detailed model for these activities in which plasminogen binds to the S100A10 subunit of AIIt and is converted to plasmin by uPA or tPA through the cleavage of the Arg-561-Val-562 bond of plasminogen. Plasmin then catalyzes the autoproteolysis of the Lys-77-Lys-78 and Lys-468-Gly-469 bonds. However, a Cys-462-Cys-541 disulfide prevents the release of the polypeptide chain (Lys-78-Lys-468) from plasmin. Thus, the ANXA2 and S100A10 subunits of AIIt catalyze the reduction of the plasmin Cys-462-Cys-541 disulfide, which allows the release of the proteolyzed polypeptide chain from the rest of the molecule. The thioredoxin system then reduces AIIt, enabling it to exert its protein reductase activity again. ecause the mutagenesis of ANXA2 Cys-334-Ser and either S100A10 Cys-61-Ser or S100A10 Cys-82-Ser inactivated the plasmin reductase activity of the isolated subunits of AIIt, and because the sum of the plasmin reactivity (AIIt) is greater than the activity exhibited by each subunit, we concluded that each subunit acts in a mutualistic symbiotic relationship to enhance its plasmin reductase activity. 

### 2.3. Mutualistic Symbiosis III—Acquisition of New Biological Activities by the Formation of the ANXA2/S100A10 Complex

Lu et al. [101] reported that the HIF-1–dependent expression of S100A10 results in the formation of a complex with ANXA2, which then interacts with the histone chaperone SPT6 (the suppressor of Ty 6 homolog *S. cerevisiae*) and histone demethylase KDM6A. Consequently, S100A10, ANXA2, SPT6, and KDM6A are recruited to OCT4 binding sites, which results in the demethylation and transcription of the genes NANOG, SOX2, and KLF4. Because the knockdown of S100A10 or ANXA2 results in the loss of this localization, the authors concluded that both S100A10 and ANXA2 are required to form this nuclear complex. Therefore, ANXA2 and S100A10 act in a mutualistic symbiotic relationship, resulting in a new function for AIIt. 

The down-regulation of the membrane protein AHNAK affects the cell membrane cytoarchitecture of epithelial cells, and plays a role in controlling the physical and mechanical properties of cell-peripheral membranes (reviewed in [102]). In vitro and in vivo experiments have shown that AHNAK interacts with AIIt [83], and that this interaction stabilizes both proteins. Specifically, the protein levels of ANXA2 and S100A10 were drastically reduced in the brains of AHNAK-KO mice [103]. Both ANXA2 and S100A10 are required for the tight association with the C-terminal domain of AHNAK [104]. More detailed studies revealed that the AHNAK binding site utilizes regions on both S100A10 and ANXA2, and NMR analysis showed that the AHNAK binding surface comprised residues from helix IV in S100A10 and the N-terminal region of ANXA2 [105]. Similarly, the immunoprecipitation of S100A10 from HEK293 cells resolved four proteins of 700, 260, 125, and 36 kDa. Mass spectrometry identified these proteins as AHNAK1, SPT6, SMARCA3 (SWI/SNF-related, matrix-associated, actin-dependent regulator of chromatin, subfamily A, member 3), and ANXA2, respectively. SMARCA3, a chromatin-remodeling factor, was shown to bind directly to AIIt. The binding site for SMARCA3 was shown to be a hydrophobic pocket formed by both S100A10 (Phe42, Ile55, Leu59, Leu75, Leu79, Ala82) and ANXA2 (Leu8, Leu11, Leu13) [106]. Therefore, ANXA2 and S100A10 act in a mutualistic symbiotic relationship resulting in the formation of a binding site for AHNAK.

The complex formed between ANXA2, S100A10 and AHNAK also interacts with dysferlin [107] (Figure 4). This large multiprotein complex facilitates the wound repair of damaged epithelial, auditory, and muscle cells upon extracellular calcium influx [105,108,109,110]. Mechanistically, it has been suggested that this dysferlin membrane repair complex, when activated by calcium, promotes the fusion of exocytosis vesicles with the inner side of the plasma membrane, resulting in the resealing of membrane tears [111,112]. The ANXA2-mediated linking of membrane surfaces under non-oxidative intracellular conditions requires ANXA2-S100A10 complex formation, because ANXA2 by itself cannot link biological membranes [80]. This suggests that the formation of the AIIt complex and its mutualistic symbiosis is required for membrane repair, and that individually ANXA2 or S100A10 cannot participate in this process.

The ANXA2/S100A10/AHNAK complex forms a complex with the L-type voltage-gated calcium channel (VGCC) [103], which plays a role in the regulation of Ca^2+^-dependent exocytosis. These channels are the major regulators that control Ca^2+^ release, and are pharmacological targets for the treatment of cardiac ischemia and hypertension [113]. A decrease in the L-type calcium influx was observed in both the glutamatergic neurons and GABAergic interneurons of AHNAK-KO mice, suggesting that that L-type calcium channels may act as effectors of the ANXA2/S100A10/AHNAK complex.

Interestingly, the S100A10 subunit of AIIt has been shown to interact with Munc13-4. Furthermore, AIIt and possibly the AIIt-AHNAK complex participate in the recruiting of Munc13-4 to Weibel–Palade body (WPB) fusion sites on the plasma membrane [114,115]. These authors suggested that Munc13-4 supports acute WPB exocytosis by tethering WPBs to the plasma membrane via AIIt.

Enlargeosomes are small cytoplasmic vesicles expressed by most cells that undergo rapid Ca^2+^-dependent exocytosis [116]. The enlargesomes function in the enlargement of the cell surface, and not for secretion into the extracellular milieu. Lorusso et al. [117] demonstrated that AIIt was required for the exocytosis of the enlargesomes, while Benaud reported the regulation of this process by the AIIt-AHNAK complex [83]. Cocucci et al. [118] showed that the exocytosis of enlargeosomes is mediated by a SNARE machinery that includes VAMP4. Baudier et al. [119] proposed that the AIIt-AHNAK complex mediated the VAMP4- and Ca^2+^ dependent exocytosis of enlargesomes.

Podocytes are specialized epithelial cells that cover the outer surfaces of glomerular capillaries (reviewed in [120]). Phospholipase A2 receptor (PLA2R) has a restricted distribution with variable low-to-medium levels within podocytes in the glomerulus [121]. It has been shown that PLA2R binds to AIIt, and that the binding site was present on S100A10 [122]. These authors suggest that AIIt plays a role in actin cytoskeleton reorganisation and tight-junction assembly in the podocytes.

ANXA2 or AIIt plays a role in the cell attachment and penetration of at least thirteen viruses, including SARS-CoV-2. ANXA2 is utilized by hepatitis C virus (HCV) and influenza A virus (IAV) during replication [123,124,125,126,127] by HPV; enterovirus 71 (EV71), respiratory syncytial virus (RSV), and cytomegalovirus (CMV) during cell attachment and penetration [128,129,130,131,132,133,134,135,136]; and by measles virus (MV) during assembly and maturation [137] (reviewed in [138]). In several cases, both the ANXA2 and S100A10 subunits have been shown to be required for this function. For example, AIIt plays a role in hepatitis B virus (HBV) intrauterine infection and mother-to-child transmission [139]. These authors showed that ANXA2 translocated from the cytoplasm to the surface of autophagosomes that contained HBV, and recruited S100A10 to form AIIt. The autophagosome then fused with MVB to form the amphisome in a Rab11-dependent process. AIIt then recruited VAMP2, and SNAP25 for membrane fusion. Because the knockdown of either ANXA2 or S100A10 affected this process, it was concluded that both of these proteins were required for HBV infection. Furthermore, Derry et al. [130] showed that the four stages of infection by cytomegalovirus were inhibited by antibodies against ANXA2 and S100A10, suggesting that AIIt is required for these processes. Woodham et al. [134] showed that S100A10 binds to amino acids 108–126 of HPV16, that AIIt coimmunoprecipitates with HPV16 at the cell surface, and that AIIt mediates HPV16 entry and infection. Dziduszko and Ozbun [135] also showed that both ANXA2 and S100A10 play a role in HPV16 entry and infection by showing that early HPV16 binding results in the translocation of AIIt to the extracellular surface. They also demonstrated that AIIt co-internalized with HPV16 and mediated intracellular trafficking. They also showed that antibodies to ANXA2 and S100A10 blocked the pre- and post-entry stages of HPV16 PSV infection. Recently, molecular in silico docking studies were performed between ANXA2 and the SARS-CoV2 spike protein using the web-based ClusPro v2.0 and the PyMol v2.4.0 programs. This analysis/evaluation showed that human ANXA2 has the potential to form stable interactions (12-hydrogen bonds and two salt bridges) with the C-terminal region of the SARS-CoV-2 spike protein. The analysis cytokine profile of COVID-19 patients suggests the release of many cytokines, which leads to lung injury, organ failure, and overall poor prognosis. Interestingly, as early as a decade ago, Fang et al. [140] employed proteomics and identified ANXA2 as a candidate auto-antigen after a SARS-CoV infection-induced cytokine storm. Moreover, they also observed the cross-reactivity between anti-SARS-Cov spike domain-2 antibodies and ANXA2. During the current COVID19 pandemic, studies have shown that anti-ANXA2 autoantibodies are elevated in severely-ill, hospitalized patients, and these upregulated levels were positively correlated with increased mortality rates [141]. Further studies will be required in order to understand whether the presence of ANXA2 auto-antibodies is a cause or effect of SARS-CoV2’s severity and pathogenesis.

Previous studies have shown that S100A10 plays a role in the translocation and/or regulation of activity of several ion channels (recently reviewed by [31]). These channels include the K+ channel, TASK-1 [142,143]; the Na+ channels, ENaC [144], Nav1.8 [145,146]; and the voltage-independent H+-gated ion channel, ASIC1a [147]. ANXA2 interacts with the cation channels such as TRPA1 [148], whereas AIIt interacts with TRPV4/5/6 [149,150,151]. Other AIIt-regulated anion channels have been reported, such as CFTR [152] and VSOR/VRAC [153]. 

Maxi-Cl, a major type of large-conductance anion channel that functions as a volume-activated Cl^−^ current, is ubiquitously expressed, and is characterized by high anion selectivity (PCl^−^/PNa^+^ of >6). This channel serves as a very efficient pathway for the release of ATP. Recently, the Maxi-Cl ion channel was shown to be regulated by AIIt, because the Maxi-Cl activity was suppressed by the cellular depletion of S100A10 or by application of a synthetic ANXA2 peptide, Ac-(1-14), which disrupts ANXA2-S100A10 complex formation [154]. The authors concluded that each subunit of AIIt plays a role in the regulation of this channel. The authors showed that ANXA2 played a role in the protein tyrosine dephosphorylation of the channel, whereas S100A10 functioned to regulate the intracellular Ca^2+^ sensitivity of the channel. Therefore, ANXA2 and S100A10 act in a mutualistic symbiotic relationship resulting in the regulation of Maxi-Cl^−^. 

TRPV5 and TRPV6 constitute the Ca^2+^ influx pathway in a variety of epithelial cells. The activites of TRPV5 and TRPV6 at the cell membrane require association with AIIt [151,155]. S100A10 binding to TRPV5 or TRPV6 was critical for channel activity and the correct targeting or retention of these channels to the plasma membrane. The role of ANXA2 was also suggested to be necessary for the proper orientation of the S100A10/TRPV5/6 complex or the lipid milieu for the channels. 

Recently, S100A10 was demonstrated to interact with the background potassium channel TASK-1 and regulate the membrane trafficking and the functionality of this channel (Girard et al., 2002). The tetrodotoxin-insensitive voltage-gated sodium channel, Nav1.8, has also been shown to bind S100A10 (Okuse et al., 2002). S100A10 promoted the translocation of Nav1.8 to the plasma membrane, thereby producing functional sodium channels. The role of ANXA2 has not been assessed for TASK-1 or Nav1.8. Therefore, it is unclear whether both ANXA2 and S100A10 are required for the activity of these ion channels.

Metabotropic glutamate receptors (mGluRs) are a sub-family of glutamate receptors. Previous studies have shown that antagonists acting on mGluR5 or mGluR2/3 exert antidepressant-like activities in mice [156]. Interestingly, an S100A10-binding motif has been identified in the cytoplasmic tail of mGluR5, and a C83Q S100A10 mutant which cannot interact with ANXA2 failed to interact with mGluR5. Therefore, the authors concluded that ANXA2 is required for the interaction of S100A10 with mGluR5 [157]. Accordingly, ANXA2 and S100A10 act in a mutualistic symbiotic relationship resulting in the regulation of mGluR5.

Mutations in CFTR, a cAMP/PKA, and the ATP-regulated Cl^-^ channel result in the disease cystic fibrosis. AIIt forms a PKA/calcineurin-dependent complex with CFTR and tethers the complex to the plasma membrane [152]. Because the forced disruption of AIIt by a peptide to the S100A10 binding site disrupts the AIIt-CFTR complex and inhibits outwardly rectifying Cl^−^ channels, the authors concluded that both subunits of AIIt were required for channel activity. Therefore, ANXA2 and S100A10 act in a mutualistic symbiotic relationship resulting in the regulation of CFTR.

Endosomes are subcellular organelles associated with the catabolism of exogenous and endogenous proteins and the down-regulation of surface receptors. They exist as three distinct cellular compartments, namely early endosomes, late endosomes, and recycling endosomes. S100A10 has been suggested to be dispensable for ANXA2 functions in the transport of endocytosed tracers from early to late endosomes. The N-terminus of ANXA2 has a membrane-binding motif (amino acids 15 to 24) that regulates its Ca^2+^-independent association with endosomes [158]. The depletion of ANXA2 prevents newly-formed multivesicular endosomes from detaching from early endosomes, and therefore regulates their transport towards late endosomes [76]. However, AIIt plays a role in the subcellular distribution of early and recycling endosomes [62,159]. In order to account for these differences, it has been suggested that calcium-dependent membrane association at the plasma membrane or along the protein recycling pathway is regulated by S100A10 binding and AIIt formation [54]. For example, S100A10 appears necessary for ANXA2 binding to the plasma membrane and the cortical actin cytoskeleton. Utilizing HPV as model cargo and BSA as pathway-specific canonical cargo in uptake experiments, Taylor et al. [136] demonstrated that S100A10 and ANXA2 are required for endosomal vesicular trafficking. They also suggested that AIIt functions in the biogenesis of multivesicular endosomes. Therefore, ANXA2 and S100A10 act in a mutualistic symbiotic relationship to regulate certain aspects of the endosomal pathway.

In conclusion, ANXA2 and S100A10 collaborate in the regulation of multiple cellular functions, including the regulation specific ion channels. In some cases, S100A10 and ANXA2 provide binding sites for an ion channel, and in other cases, each subunit may play distinct but essential regulatory roles. In some cases, ANXA2 may tether the S100A10-ligand/channel complex to the plasma membrane. The ANXA2-S100A10-AHNAK complex also plays key roles in the regulation of L-type voltage-gated calcium channel and in the membrane repair process.

## 3. Unresolved Functions

Svenningsson initially reported the regulation of the cell surface expression of specific serotonin receptors by S100A10. He observed that the third intracellular loop of serotonin receptors, 5-HT1BR, 5-HT1DR, and 5-HT4R, interacted with S100A10, and that S100A10 also increased the surface levels of 5-HT1BR and 5-HT4R in transfected cell lines [46,160]. However, it is unclear if ANXA2 is also involved in these activities of S100A10. It is also unclear if ANXA2, S100A10, or AIIt plays a role in the cell attachment and penetration of the thirteen viruses that are dependent on ANXA2, although several of these viruses have been shown to require both proteins.

## Figures and Tables

**Figure 1 biomolecules-11-01849-f001:**
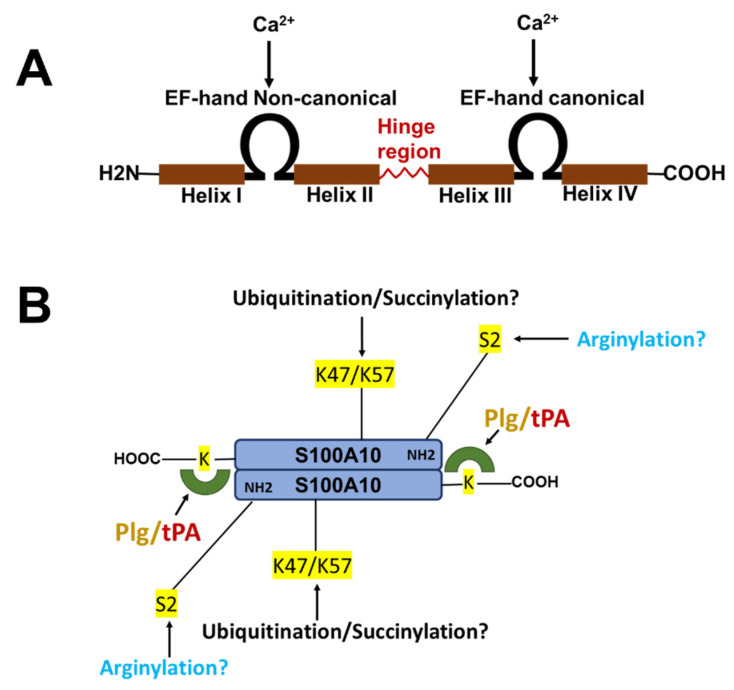
Structure of S100 proteins. (**A**) The structure of a typical S100 protein. (**B**) The structure and key regulatory sites of S100A10. Shown are the sites for ubiquitylation, arginylation and succinylation.

**Figure 2 biomolecules-11-01849-f002:**
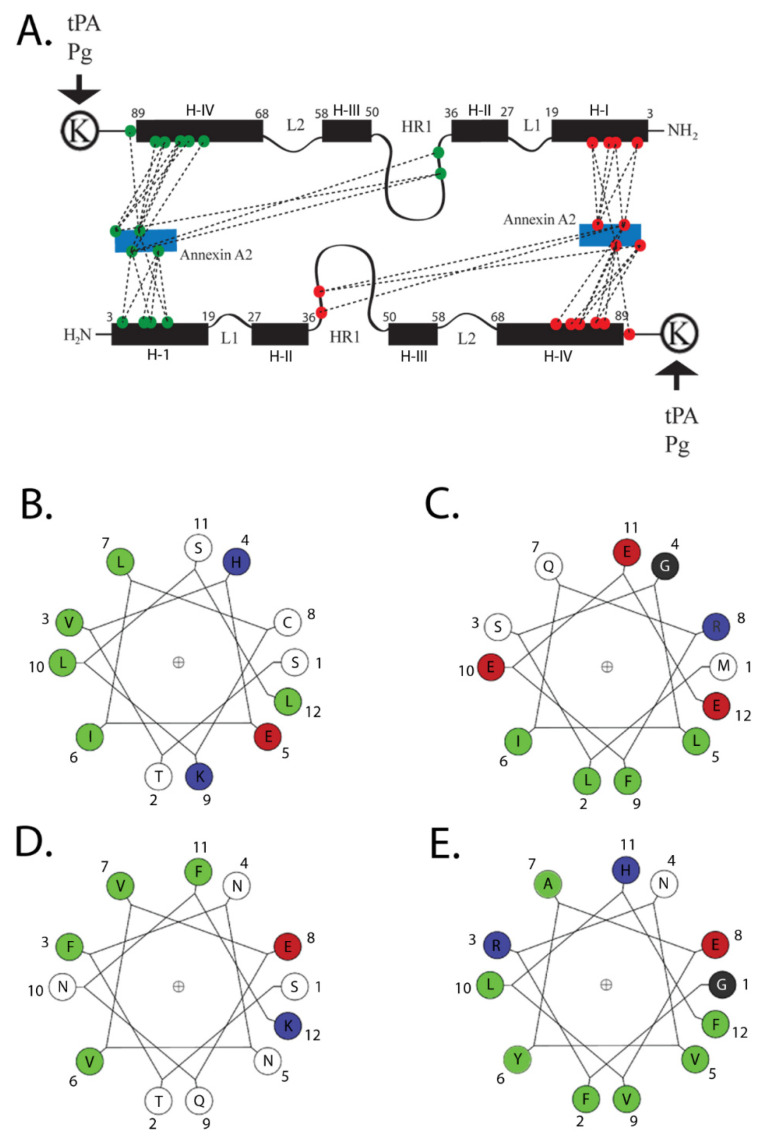
Structure of AIIt. The numbering of the amino acids represents the position of the amino acids in the wheel. Colors are—green, hydrophobic; red, acidic; blue, basic; black, neutral. A cartoon of the association of S100A10 with its two primary ligands, ANXA2 and plasminogen, is presented in (**A**). The figure illustrates the structure of S100A10 and the association of S100A10 with the amino-terminus of ANXA2 and with plasminogen. Each S100A10 monomer is composed of four α-helical domains: H-I, H-II, H-III and H-IV. Separating the H-I and H-II helical regions is a loop, L1. The H-III and H-IV are separated by a second loop (L2). The H-II and H-III are connected by a flexible linker or hinge region (HR1). The points of interaction between the amino-terminus of ANXA2 and S100A10 are quite extensive, and four hydrophobic amino acids of the amino terminus of ANXA2 (V3, I6, L7 and L10) form seven points of contact with helix H-I of one monomer, two points of contact with the hinge region, and nine points of contact with helix H-IV of the other monomer, for a total of nineteen points of contact with S100A10. (**B**) The helical wheel projections for the S100A10-binding site for ANXA2. (**C**) Bluetongue virus, NS3; (**D**) DLC1; and (**E**) TASK-1 are presented. The S100A10-binding region of these ligands consists of an amphipathic α-helix in which hydrophobic residues form a binding site on one side of the helix. The program for helical wheel projections was obtained from http://lbqp.unb.br › NetWheels/, accessed on 5 November 2021. The numbering of the amino acids represents the position of the amino acids in the wheel. This cartoon and figure legend was originally published in Madureira, O’Connell, Surette, Miller, and Waisman 2012, ‘The Biochemistry and Regulation of S100A10: A Multifunctional Plasminogen Receptor Involved in Oncogenesis. J Biomed Biotechnol 2012:353687’.

**Figure 3 biomolecules-11-01849-f003:**
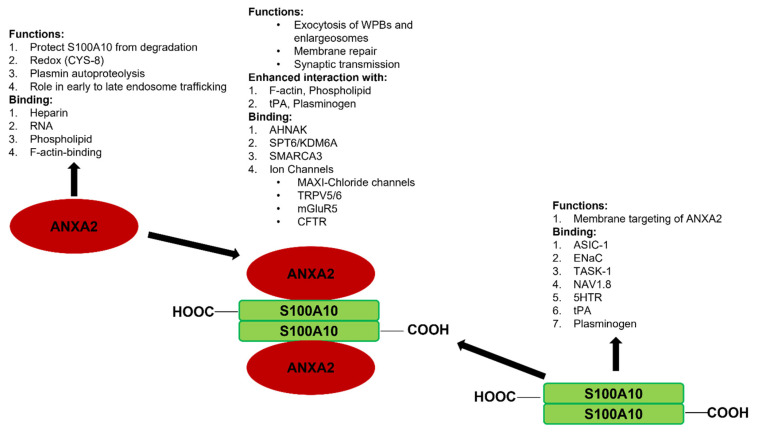
Synopsis of the mutualistic symbiotic relationship between S100A10 and ANXA2. Annexin A2 is a monomeric Ca^2+^-binding protein that binds F-actin, phospholipid, RNA, and specific polysaccharides, such as heparin. S100A10 does not bind Ca^2+^, but binds tPA, plasminogen, certain plasma membrane ion channels, and neurotransmitter receptors. ANXA2 and S100A10 form a tight 2:2 heterotetrameric complex, which is called AIIt. The formation of AIIt results in the enhancement of many of the functions of each subunit. For example, ANXA2 enhances the ability of S100A10 to bind plasminogen and facilitate the tPA-dependent conversion of plasminogen to plasmin, whereas S100A10 enhances the interaction of ANXA2 with phospholipids. The formation of AIIt permits the interaction of the complex with new binding partners such as certain ion channels, dysferlin, and the structural scaffold protein AHNAK.

**Figure 4 biomolecules-11-01849-f004:**
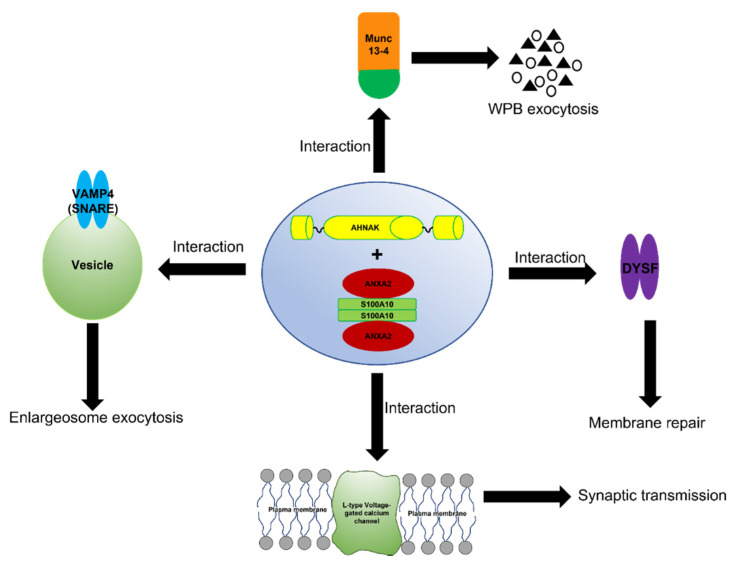
Schematic representation of the function of the AIIt-AHNAK complex. The formation of the ANXA2/S100A10/AHNAK (2:2:1) complex forms a key hub for the regulation of several important biological functions such as the exocytosis of enlargesomes, the regulation of the L-type voltage-gated calcium channel (VGCC), and possibly the MUNC-13-dependent regulation of the exocytosis of Weibel–Palade bodies (WPB).

## Data Availability

Not applicable.

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
