# Peer review of "The Annexin A2/S100A10 Complex: The Mutualistic Symbiosis of Two Distinct Proteins"

_biomolecules, 2021, doi:10.3390/biom11121849_

Round 1

Reviewer 1 Report

This review provides a comprehensive overview of the symbiotic functions of AnnexinA2 and S100A10 proteins. The review is very detailed and enjoyable to read.

To help the readers not familiar with the subject, I would only suggest that the authors include a schematic and colourful representation (with legend) of the multiple interactions of the Anexin2-S100A10 complex in assembly of components of Ca2+-dependent pathways (i.e. L-type voltage-gated calcium channel) and exocytosis. Few references on the functional role of the AHNAK1/S100A10/Annexin2 complex in exocytosis might also be interesting to cite to complete the picture. When anchored to the plasma membrane, AHNAK1 scaffolds L-type voltage-gated calcium channels (VGCCs) thereby regulating downstream Ca2+-dependent pathways and exocytosis (Jin, J. et al. Ahnak scaffolds p11/Anxa2 complex and L-type voltage-gated calcium channel and modulates depressive behavior. Mol Psychiatry 2019, 10.1038/s41380-019-0371), and recruits phospholipase C and Ca2+-dependent PKC-α required for PI(4,5)P2 synthesis (Lee, I.H. et al. AHNAK-mediated activation of phospholipase C-gamma1 through protein kinase C. J Biol Chem 2004, 279, 26645-26653). (PI(4,5)P2) is a crucial component of the plasma membrane forming microdomains required for efficient SNARE-mediated exocytosis. At the plasma membrane, the AHNAK1/S100A10/Annexin2 complex may evolves into vesicles called "enlargosomes” that are rapidly exocytosed in a SNARE- and Ca2+-dependant manner (Lorusso, A. et al. Annexin2 coating the surface of enlargeosomes is needed for their regulated exocytosis. EMBO J 2006, 25, 5443-5456; Cocucci, E.; Racchetti, G.; Rupnik, M.; Meldolesi, J. The regulated exocytosis of enlargeosomes is mediated by a SNARE machinery that includes VAMP4. J Cell Sci 2008, 121, 2983-2991; Reviewed in Baudier,J. and Gentil, B. Biomolecule 2020, 10, 843). 

Author Response

To help the readers not familiar with the subject, I would only suggest that the authors include a schematic and colourful representation (with legend) of the multiple interactions of the Anexin2-S100A10 complex in assembly of components of Ca2+-dependent pathways (i.e. L-type voltage-gated calcium channel) and exocytosis. Few references on the functional role of the AHNAK1/S100A10/Annexin2 complex in exocytosis might also be interesting to cite to complete the picture. When anchored to the plasma membrane, AHNAK1 scaffolds L-type voltage-gated calcium channels (VGCCs) thereby regulating downstream Ca2+-dependent pathways and exocytosis (Jin, J. et al. Ahnak scaffolds p11/Anxa2 complex and L-type voltage-gated calcium channel and modulates depressive behavior. Mol Psychiatry 2019, 10.1038/s41380-019-0371), and recruits phospholipase C and Ca2+-dependent PKC-α required for PI(4,5)P2 synthesis (Lee, I.H. et al. AHNAK-mediated activation of phospholipase C-gamma1 through protein kinase C. J Biol Chem 2004, 279, 26645-26653). (PI(4,5)P2) is a crucial component of the plasma membrane forming microdomains required for efficient SNARE-mediated exocytosis. At the plasma membrane, the AHNAK1/S100A10/Annexin2 complex may evolves into vesicles called "enlargosomes” that are rapidly exocytosed in a SNARE- and Ca2+-dependant manner (Lorusso, A. et al. Annexin2 coating the surface of enlargeosomes is needed for their regulated exocytosis. EMBO J 2006, 25, 5443-5456; Cocucci, E.; Racchetti, G.; Rupnik, M.; Meldolesi, J. The regulated exocytosis of enlargeosomes is mediated by a SNARE machinery that includes VAMP4. J Cell Sci 2008, 121, 2983-2991; Reviewed in Baudier,J. and Gentil, B. Biomolecule 2020, 10, 843). 

We have added . --The complex formed between ANXA2, S100A10 and AHNAK also interacts with dysferlin [110]. This large multiprotein complex facilitates wound repair of damaged epithelial, auditory, and muscle cells upon extracellular calcium influx [108,111–113]. Mechanistically, it has been suggested that this dysferlin membrane repair complex, when activated by calcium, promotes the fusion of exocytosis vesicles with the inner side of the plasma membrane resulting in the resealing of membrane tears [114,115]. The ANXA2-mediated linking of membrane surfaces under non-oxidative intracellular conditions requires ANXA2-S100A10 complex formation, since ANXA2 by itself cannot link biological membranes [83]. This suggests that the formation of the AIIt complex and their mutualistic symbiosis is required for membrane repair and that individually ANXA2 or S100A10 cannot participate in this process.

The ANXA2/S100A10/AHNAK complex forms a complex with the L-type voltage-gated calcium channel (VGCC) [106] which plays a role in the regulation of Ca2+-dependent exocytosis. These channels are the major regulators that control Ca2+ release and are pharmacological targets for the treatment of cardiac ischemia and hypertension [116]. Interestingly, the S100A10 subunit of AIIt has been shown to interact with Munc13-4. Furthermore, AIIt participates in recruiting Munc13-4 to Weibel-Palade bodies (WPB) fusion sites on the plasma membrane [117]. These authors suggested that Munc13-4 supports acute WPB exocytosis by tethering WPBs to the plasma membrane via AIIt.

Interestingly, the S100A10 subunit of AIIt has been shown to interact with Munc13-4. Furthermore, AIIt and possibly the AIIt-AHNAK complex, participates in recruiting Munc13-4 to Weibel-Palade bodies (WPB) fusion sites on the plasma membrane [117,118]. These authors suggested that Munc13-4 supports acute WPB exocytosis by tethering WPBs to the plasma membrane via AIIt.

Enlargeosomes are small cytoplasmic vesicles, expressed by most cells that undergo rapid Ca2+-dependent exocytosis [119]. The enlargesomes function in the enlargement of the cell surface and not for secretion into the extracellular milieu. Lorusso et al. [120] demonstrated that AIIt was required for the exocytosis of the enlargesomes while Benaud reported the regulation of this process by the AIIt-AHNAK complex [86]. Cocucci et al. [121] showed that exocytosis of enlargeosomes is mediated by a SNARE machinery that includes VAMP4. Baudier et al. [122] proposed that the AIIt-AHNAK complex mediated the VAMP4- and Ca2+dependent exocytosis of enlargesomes.

Podocytes are specialized epithelial cells that cover the outer surfaces of glomerular capillaries (reviewed in [123]). Phospholipase A2 receptor (PLA2R) has a restricted distribution with variable low to medium levels within podocytes in the glomerulus [124]. It has been shown that PLA2R binds to AIIt and that the binding site was present on S100A10 [125]. These authors suggest that AIIt plays a role in actin cytoskeleton reorganisation and tight junction assembly in the podocytes.

However while AHNAK recruits phospholipase C and Ca2+-dependent PKC-α required for PI(4,5)P2 synthesis (Lee, I.H. et al. AHNAK-mediated activation of phospholipase C-gamma1 through protein kinase C. J Biol Chem 2004, 279, 26645-26653) it has not been demonstrated that AIIt is associated with this complex. I think it is very likely but until this is directly demonstrated I do not think it appropriate to shown an AIIt/AHNAK complex associating with these proteins. Also Munc13-4 supports acute WPB exocytosis by tethering WPBs to the plasma membrane via AnxA2-S100A10-no mention of AHNAK in this paper but I added this to the figure with a question mark.

Reviewer 2 Report

The Review entitled “The Annexin A2/S100A10 Complex: the mutualistic symbiosis of two distinct proteins” proposed by David Waisman and collaborators is well written and of interest, for either specialists of annexins and S100 proteins or non-specialists. The way the authors address the biological relevance of the ANXA2/S100A10 complex, by focusing on the symbiotic relationship of both partners, is particularly relevant and original. The Review nevertheless suffers from the absence of illustrations (images and scheme) to support statements.

Major comments:

  • p.1, line 44: description of the structure of S100 proteins requires a figure.
  • The ANX abbreviation has to be introduced, probably p.1, line 54: “interact with several annexin (ANX) proteins.”
  • p.2, line 35: description of the structure of ANXA2 and the complex it forms with S100A10 requires a figure, particularly this part: “The N-terminal domain (residues 2-33) is an intrinsically disordered region. The first twelve residues form an amphipathic α-helix and the hydrophobic face of this helix, specifically Val3, Ile6, Leu7, and Leu10 of S100A10, interacts with a hydrophobic surface on the S100A10 dimer [18,23–25]. These four hydrophobic amino acids of the N-terminal domain of ANXA2 form seven points of contact with helix H-I of one monomer, two points of contact with the hinge region, and nine points of contact with helix H-IV of the other monomer, for a total of nineteen points of contact between ANXA2 and S100A10 (reviewed in [12,21]. This region also bears two important phosphorylation sites, Tyr23 and Ser25, which are targets of SRC kinase and protein kinase C, respectively [7,17,26]”.
  • p.2, line 10: There is some redundancy in the two following sentences “The ANXA1, ANXA2, ANXA5, ANXA6, and ANXA11 interact with S100 proteins [9]. S100A1, S100A4, S100A6, S100A10, S100A11, S100A12, and S100B can interact with annexins, whereas ANXA1, ANXA2, ANXA5, ANXA6, and ANXA11 can interact with these S100 proteins.”
  • The new biological functions brought by the association of ANXA2 to S100A10 may be displayed in the middle panel of Figure 1.
  • p.4, line 26: “(Ikebuchi and Waisman, 1990)” is incorrectly referenced.
  • p.5, line 14: “(Drust and Creutz, 1988)” is incorrectly referenced.
  • Many recent studies have shown that ANXA2, S100 proteins, and AHNAK are involved in cell membrane repair, among the unresolved functions the authors may mention the putative role play by AIIt in this physiological process.

Minor typographic errors:

  • p.1, line 40: “Other members (S100A11P, S100B, S100G, S100P, and S100Z) map to different”…“P” in “S100A11P” has to be removed
  • p.1, line 52: “but retains the “active” conformation of a calcium-bound S100 protein [7]._” The underscore at the end of the sentence has to be removed.
  • p.5, line 38: “However,, a proteolytically processed form…” a coma has to be removed after “however”
  • p.7, line 33: “The analysis cytokine profile of COVD-19 patients”, COVID-19 has to be corrected
  • p.9, line 1: an extra space has to be removed after “ANXA2” in the sentence “In conclusion, ANXA2 and S100A10 collaborate…”

Author Response

The Review entitled “The Annexin A2/S100A10 Complex: the mutualistic symbiosis of two distinct proteins” proposed by David Waisman and collaborators is well written and of interest, for either specialists of annexins and S100 proteins or non-specialists. The way the authors address the biological relevance of the ANXA2/S100A10 complex, by focusing on the symbiotic relationship of both partners, is particularly relevant and original. The Review nevertheless suffers from the absence of illustrations (images and scheme) to support statements.

Thank you for your positive comments. As per your request, we have added several figures.

Major comments:

  • p.1, line 44: description of the structure of S100 proteins requires a figure.

We have added a new figure--as figure 1.

  • The ANX abbreviation has to be introduced, probably p.1, line 54: “interact with several annexin (ANX) proteins.”

This has been corrected

  • p.2, line 35: description of the structure of ANXA2 and the complex it forms with S100A10 requires a figure, particularly this part: “The N-terminal domain (residues 2-33) is an intrinsically disordered region. The first twelve residues form an amphipathic α-helix and the hydrophobic face of this helix, specifically Val3, Ile6, Leu7, and Leu10 of S100A10, interacts with a hydrophobic surface on the S100A10 dimer [18,23–25]. These four hydrophobic amino acids of the N-terminal domain of ANXA2 form seven points of contact with helix H-I of one monomer, two points of contact with the hinge region, and nine points of contact with helix H-IV of the other monomer, for a total of nineteen points of contact between ANXA2 and S100A10 (reviewed in [12,21]. This region also bears two important phosphorylation sites, Tyr23 and Ser25, which are targets of SRC kinase and protein kinase C, respectively [7,17,26]”.

We have added this figure as figure 2. The figure was originally published in our 2012 review.

  • p.2, line 10: There is some redundancy in the two following sentences “The ANXA1, ANXA2, ANXA5, ANXA6, and ANXA11 interact with S100 proteins [9]. S100A1, S100A4, S100A6, S100A10, S100A11, S100A12, and S100B can interact with annexins, whereas ANXA1, ANXA2, ANXA5, ANXA6, and ANXA11 can interact with these S100 proteins.”

We have removed “whereas ANXA1, ANXA2, ANXA5, ANXA6, and ANXA11 can interact with these S100 proteins.”

  • The new biological functions brought by the association of ANXA2 to S100A10 may be displayed in the middle panel of Figure 1.--done

  • p.4, line 26: “(Ikebuchi and Waisman, 1990)” is incorrectly referenced.

corrected

  • p.5, line 14: “(Drust and Creutz, 1988)” is incorrectly referenced.

corrected

  • Many recent studies have shown that ANXA2, S100 proteins, and AHNAK are involved in cell membrane repair, among the unresolved functions the authors may mention the putative role play by AIIt in this physiological process.

Agreed--we have added a paragraph on membrane repair.

Minor typographic errors:

  • p.1, line 40: “Other members (S100A11P, S100B, S100G, S100P, and S100Z) map to different”…“P” in “S100A11P” has to be removed

done

  • p.1, line 52: “but retains the “active” conformation of a calcium-bound S100 protein [7]._” The underscore at the end of the sentence has to be removed.

done

  • p.5, line 38: “However,, a proteolytically processed form…” a coma has to be removed after “however”

done

  • p.7, line 33: “The analysis cytokine profile of COVD-19 patients”, COVID-19 has to be corrected

done

  • p.9, line 1: an extra space has to be removed after “ANXA2” in the sentence “In conclusion, ANXA2 and S100A10 collaborate…”

done

Thank you for your meticulous reading of the manuscript.

Reviewer 3 Report

This is an interesting review work involving S100A10 and ANXA2. Here authors discussed the functional aspect of the S100A10-ANXA2 complex. According to the reviewed work, the author suggests that there exists a mutualistic symbiotic relationship between these two proteins with an important influence on biological activity. It S100 family of proteins are Ca2+ binding proteins with the exception of S100A10. It has also been discussed that ANXA2 and S100A10 play an important role in the regulation of specific ion channels with the availability of binding sites for a channel.

Here are a few important concerns that authors should consider addressing:

  1. The functions of these molecules are intrinsically related to their dynamics. It would be important to discuss how the dynamical characteristic of S100A10, a non-calcium binding protein, varies with respect to its calcium-binding family members as they function.  
  2. How do the structural dynamics of other calcium-binding proteins with helix-loop-helix arrangements, such as Calmodulin, compare with the S100 family? Are they similar with respect to the E-F motif dynamics?
  3. Authors should consider discussing Brenchley, P.E. et al. work on A2- S100A10 complex in human podocytes in this context.

Author Response

This is an interesting review work involving S100A10 and ANXA2. Here authors discussed the functional aspect of the S100A10-ANXA2 complex. According to the reviewed work, the author suggests that there exists a mutualistic symbiotic relationship between these two proteins with an important influence on biological activity. It S100 family of proteins are Ca2+ binding proteins with the exception of S100A10. It has also been discussed that ANXA2 and S100A10 play an important role in the regulation of specific ion channels with the availability of binding sites for a channel.

Here are a few important concerns that authors should consider addressing:

  1. The functions of these molecules are intrinsically related to their dynamics. It would be important to discuss how the dynamical characteristic of S100A10, a non-calcium binding protein, varies with respect to its calcium-binding family members as they function.  

Thank you for your comments. Respectfully, the review looks at how the combination of S100A10 and ANXA2 contributes to novel functions for the complex ie. ANXA2 changes the dynamical characteristics of S100A10 (and visa versa) and not how S100A10  dynamical characteristics are unique among the S100 family. As a compromise, we have added a comparison between S100A4 and S100A10 since both are plasminogen receptors and S100A4 but not S100A10 activates plasminogen in a Ca2+ dependent manner. We have also added-The S100 proteins are typically dimeric and have two specialized binding regions that are either identical or slightly different in the case of homo- and heterodimers, respectively [21]. These binding regions act as anchor points for their ligands but are not surface exposed in the absence of Ca2+. They become accessible when Ca2+ binds to the EF-hands of the S100 proteins and induces a conformational rearrangement in them. These binding regions in S100A10 are surface exposed in the presence or absence of Ca2+. Between the binding pockets of the S100 proteins there exists a shallow groove. This region is located at the dimeric interface. The shallow groove is less specific with regard to the binding partners that interact with this site (reviewed in [22]. Two identical binding proteins can occupy these sites  as exemplified by the ANXA2–S100A10 complex [23]. In contrast, in some cases a single binding sequence utilizes both sites and can also utilize the groove to various extents [22]. For example, two molecules of S100A4 bind to a single ANXA2 molecule ie. one molecule of ANXA2 occupies both binding pockets of the S100A4 dimer whereas only one molecule of ANXA2 occupies only one site on each S100A10 molecule in the dimer. It has been suggested that in order to be able to bind a wide variety of ligands without optimal binding motifs, S100 proteins may have developed relatively permissive binding sites leading to more flexible interactions [18,22].

  1. How do the structural dynamics of other calcium-binding proteins with helix-loop-helix arrangements, such as Calmodulin, compare with the S100 family? Are they similar with respect to the E-F motif dynamics?

We have included a diagram of the structure of S100A10 that shows the pseudoEF  and EF type binding sites. We have included a discussion in the text. ---Specifically, the S100 proteins possess an S100-specific EF hand at the N-terminus and a canonical EF hand at the C-terminus. In all of the S100 proteins except S100A10, the EF hands bind Ca2+. These EF hands are not identical; the carboxyl-terminal EF hand Ca2+-binding loop contains the classical 12-amino-acid long Ca2+-binding motif that is common to all EF hand Ca2+-binding proteins such as parvalbumin and calmodulin. In this motif, Ca2+-binding occurs via acidic side chains that comprise the sequence DXDGDGTIXXXE. However, the N-terminal EF hand motif is a 14-amino-acid long Ca2+-binding loop which is characteristic of all S100 proteins and is referred to as the S100 specific or pseudo EF domain. This motif binds Ca2+ via backbone carbonyl groups and one carboxylate side group contributed by glutamic acid (reviewed in [7–9]). All the S100 proteins bind calcium, except S100A10, which has lost its ability to bind calcium due to substitutions in its calcium-binding loop but retains the “active” conformation of a calcium-bound S100 protein [10]. S100A10 has lost its ability to bind calcium due to substitutions in its calcium-binding loop but retains the activated structure of a calcium-bound S100 protein [11]

  1. Authors should consider discussing Brenchley, P.E. et al. work on A2- S100A10 complex in human podocytes in this context.

Agreed--We have added a paragraph. --Podocytes are specialized epithelial cells that cover the outer surfaces of glomerular capillaries (reviewed in [111]). Phospholipase A2 receptor (PLA2R) has a restricted distribution with variable low to medium levels within podocytes in the glomerulus [112]. It has been shown that PLA2R binds to AIIt and that the binding site was present on S100A10 [113]. These authors suggest that AIIt plays a role in actin cytoskeleton reorganisation and tight junction assembly in the podocytes.

Round 2

Reviewer 3 Report

The authors have adequately addressed concerns with

  1. Added discussion with an elaborate explanation.
  2. New figure and more clear explanation and comparison.
  3. Further examination of more relevant literature and added discussion panel.

It now is a much more complete form of the review than the earlier version, and I am ready to accept the current document.